# MicroRNA Profile of Human Small Intestinal Tumors Compared to Colorectal Tumors

**DOI:** 10.3390/jcm11092604

**Published:** 2022-05-06

**Authors:** Yoshihito Nakagawa, Yukihiro Akao, Hiromi Yamashita, Tomomitsu Tahara, Kohei Funasaka, Mitsuo Nagasaka, Teiji Kuzuya, Ryoji Miyahara, Senju Hashimoto, Tomoyuki Shibata, Yoshiki Hirooka

**Affiliations:** 1Department of Gastroenterology and Hepatology, School of Medicine, Fujita Health University, Kutsukake-cho, Toyoake 470-1192, Aichi, Japan; shokakan@fujita-hu.ac.jp (H.Y.); tomomiccyu@yahoo.co.jp (T.T.); k-funa@med.nagoya-u.ac.jp (K.F.); nmitsu@fujita-hu.ac.jp (M.N.); teiji.kuzuya@fujita-hu.ac.jp (T.K.); ryoji.miyahara@fujita-hu.ac.jp (R.M.); hsenju@fujita-hu.ac.jp (S.H.); shibat03@fujita-hu.ac.jp (T.S.); yoshiki.hirooka@fujita-hu.ac.jp (Y.H.); 2The United Graduate School of Drug Discovery and Medical Information Sciences, Gifu University, Gifu 501-1193, Gifu, Japan; yakao@gifu-u.ac.jp

**Keywords:** onco-related microRNA, miR-143, miR-145, miR-31, small intestinal adenoma and adenocarcinoma, colorectal adenoma and adenocarcinoma, carcinogenesis

## Abstract

Small intestinal tumors (adenoma and adenocarcinoma, SIT) are rare, and their microRNA (miRNA) expression profiles have not been established. Previously, we reported a relationship between miRNA expression profiles and the development, growth, morphology, and anticancer drug resistance of colorectal tumors. Here, we demonstrate that the miRNA expression profile of SIT is significantly different from those of tumors of the colon. We compared the onco-related miRNA expression profiles of SIT and colorectal tumors and found them to be different from each other. The expressions of miR-143 and miR-145 were frequently downregulated in SIT and colorectal tumors but not in sessile serrated adenoma/polyp tumors. The profiles of SIT and colorectal carcinomas of miR-7, miR-21, and miR-34a were considerably different. Upregulation of miR-31 expression was not found in any SIT cases. Our data suggested that miR-143 and miR-145 might act as anti-oncomirs common to adenocarcinoma of the small intestine, similar to those of colorectal adenoma and other cancers. However, the expression profiles of the other miRNAs of SIT were significantly different from those of colorectal tumors. These findings contribute useful insights into the tumor development and diagnosis of SIT.

## 1. Introduction

Endoscopy of the small intestines has improved remarkably [1,2,3,4], and balloon-assisted and/or capsule endoscopy is now used to treat small intestinal diseases. However, balloon-assisted endoscopy is more complex and causes more pain to patients than gastrointestinal or colorectal endoscopy. In addition, capsule endoscopy provides poor resolution due to the influence of food residues, which also interferes with biopsy.

Although the area of the small intestinal membrane is larger than that of the stomach or colorectum, small intestinal tumors (SITs) are very rare compared to cancers of the stomach and colorectum. Therefore, SIT is often difficult to treat when found [5,6,7]. The reasons explaining this incredibly low prevalence of SIT are unclear, but the genetic abnormalities of SIT are significantly different from those of gastric cancer or colorectal tumors [6,7]. For example, the frequencies of APC and CDKN2A gene abnormalities in small intestinal adenocarcinoma are lower than in colorectal cancer [6], but KRAS gene abnormalities are observed as frequently as in colorectal tumors [7]. BRAF gene abnormalities in small intestinal adenocarcinoma occur a little less than 10% of the time, but V600E mutations in small intestinal adenocarcinoma occur under 1% of cases, unlike in colorectal cancer [6].

MicroRNAs (miRNAs) are endogenous noncoding RNAs of approximately 22–25 nucleotides in length that inhibit the translation of mRNAs in a sequence-specific manner [8,9,10]. In humans, 2693 mature miRNAs have already been identified [11], and many onco-related miRNA have been reported to contribute to development, proliferation, cell death, and drug resistance in various cancers [12,13,14,15,16]. Moreover, miRNAs provide stability to messenger RNAs [17]. We previously reported a relationship between miR-143, miR-145, miR-7, miR-21, and miR-34a expression profiles and the development, growth, and morphology of colorectal and gastric tumors [18,19,20,21]. Additionally, we previously reported a relationship between miR-31 and anticancer drug resistance of colorectal tumors [22].

Because the genetic abnormalities of small intestinal adenocarcinoma are significantly different from those of colorectal cancer, it is possible that the miRNA expression profile of SIT is also different from that of colorectal tumors. In this study, we examined the miRNA expression profile in SIT of six onco-related and/or drug-resistance-related miRNAs of colorectal tumors to see if this was accurate.

## 2. Materials and Methods

### 2.1. Patients and Tissue Preparation

We examined the miRNA expression levels in 27 sporadic SIT cases. The first diagnosis of duodenal tumors was made by esophagogastroduodenoscopy. The first diagnosis of tumors in the jejunum was made with double-balloon endoscopy. The SIT patients consisted of 20 men and 7 women, aged 41–88 (median 70), and the tumor sizes were 4–60 mm (median 30 mm). SITs consisted of two tubular adenomas and 25 adenocarcinomas (24 well or moderately differentiated adenocarcinomas and 1 poorly differentiated adenocarcinoma) without ampullary duodenal tumors. The depths of the adenocarcinomas were as follows: 11 carcinomas in situ, 1 in the submucosa, none in the muscularis propria, 4 in the subserosa, 5 in the serosa, and 4 over serosa. The sites of SIT were as follows: 6 in the 1st, 12 in the 2nd, 3 in the 3rd part of the duodenum, and 6 in the jejunum. In this study, there were no samples from the ileum. There were also no samples from familial adenomatous polyposis (FAP) or inflammatory bowel disease. Hereditary nonpolyposis colorectal carcinoma (HNPCC) was suspected in one case.

We further examined the chosen miRNA expression levels in 172 sporadic colorectal tumors (59 tubular or tubulo-villous adenoma, 71 well- or moderately differentiated adenocarcinomas (25 carcinomas in situ or carcinomas invading submucosa, and 46 advanced carcinomas), 26 adenomas from FAP, and 16 sessile serrated adenomas/polyps (SSA/P)). The ages of colorectal tumor patients were between 18 and 88 (median 66) years, and there were 104 men and 68 women. The sizes of the colorectal tumors were between 3 and 130 (median 16) mm, and with respect to the sites, there were 62 right and 110 left.

Seven SIT and all human colorectal tumor samples were obtained in a fresh state from patients who had undergone a direct biopsy for diagnosis or surgery for resection of colorectal tumors at Fujita Health University Hospital (Aichi, Japan), Saiseikai Ibaraki hospital (Osaka, Japan), Osaka Medical Pharmaceutical University Hospital (Osaka, Japan), or Kyoritsu General Hospital (Aichi, Japan) between 2002 and 2021. Twenty SIT samples were obtained in formalin-fixed, paraffin-embedded (FFPE) samples from patients who had undergone surgery for resection at Fujita Health University Hospital. Two pathologists diagnosed each sample based on the Japanese Classification of Colorectal Carcinoma (9th edition) [23]. Informed consent in writing was obtained from each patient at each hospital. Additionally, the protocol was approved by the Ethics Committee of Fujita Health University Hospital. We analyzed the expression levels of identified miRNAs in these tumor tissues in comparison with adjacent normal small intestine or colon tissues in the same patients.

### 2.2. RNA Isolation and Quantitative Real-Time PCR

Total RNA was isolated from fresh tissues by TRIzol containing phenol/guanidium isothiocyanate and then treated with DNase I [18,20]. Total RNA was isolated from FFPE tissues by use of the NucleoSpin^®^ totalRNA FFPE (Takara, Shiga, Japan) kit according to the manufacturer’s instructions.

To examine the expression levels of miRNAs, we performed TaqMan^®^ MicroRNA Assays using a real-time PCR apparatus (Thermo Fisher Scientific, Waltham, MA, USA) [18,19,20,21,22]. We examined the expression levels of tumor miRNAs compared with those of the paired normal samples in a blinded fashion. The threshold cycle (Ct) was defined as the fractional cycle number at which the fluorescence passes a fixed threshold. The range of Ct values of these miRNAs in colorectal cancer was from 18 to 40. The levels of miRNAs in each tissue were measured and normalized to those of U6, which was used as an internal control [18,20]. The relative expression levels were calculated by the ΔΔCt method. The relative expression level in normal tissue was set as 1. The tumor/non-tumor ratio of each miRNA expressed in the samples was determined and depicted using box-and-whisker plots.

### 2.3. Statistics and Data Analysis

The expression levels in tumors were designated as downregulated when the fold change from the expression in the non-tumorous tissues was 0.67 or as upregulated when the fold change was 1.50, as determined from the results of linear discriminant analysis of miR143, miR-145, miR-7, miR-21, and miR-34a expression patterns from pairs of tumors and non-tumorous tissues [18,19,20]. In experiments on clinical samples, an expression level > 10.0 was designated as upregulation, for which fold changes were obtained from the results of linear discriminant analysis of miR-31 expression patterns from pairs of tumors and non-tumorous tissues [22]. Each examination was performed in triplicate, and the average was used in this study. Statistical differences in miRNA levels were evaluated using Pearson’s χ^2^ test or Fisher’s extract test for differences between two groups. A *p* value of 0.05 was considered to be significant. All calculations were performed using Bell Curve for Excel (version 2; SSRI Co., Ltd., Tokyo, Japan).

## 3. Results

### 3.1. miRNA Profiles of Human Small Intestinal Tumors

We examined the differences in miRNA profiles between FFPE and fresh samples in the same sample of two cases (Appendix A), and no difference between the FFPE and fresh samples was found. We previously reported that miR-143 and miR-145 are transcribed from the same primary miRNA at 5q33 [24], so their expression profiles were found to be similar in many studies [18,19,20,21,22].

The expression levels of these miRNAs were not related to site or depth (Table 1) or to sex, age, or the size of SIT. The expression levels of miR-143 and miR-145 were downregulated in almost 60% of SIT cases. The expression levels of miR-7 and miR-21 were upregulated in 29.6% and 44.4% of SIT cases, respectively. The expression levels of miR-34a were downregulated in 22.2% of SIT cases. The expression levels of miR31 were unchanged in all SIT cases (Table 1).

### 3.2. miRNA Profiles of Human Colorectal Tumors

miR-143 and miR-145 were frequently downregulated in colorectal tumors but not in SSA/P. The expression levels of miR-7 were upregulated in colorectal adenocarcinoma and SSA/P. The expression frequencies of miR-21 and miR-34a were similar to those of SIT and colorectal tumors. The expression levels of miR31 were upregulated in advanced cancer and SSA/P (Table 1). As in previous reports [18,19], there was no difference in miRNA expression profile by sex, age, site, or size of colorectal tumors.

### 3.3. miRNA Profiles of Human Small Intestinal Tumors Compared to Colorectal Tumors

Subsequently, we compared the miRNA expression profiles of SIT and colorectal tumors (Figure 1 and Figure 2). Except in SSA/P, miR-143 and miR-145 were frequently decreased in SIT and colorectal tumors (Table 1, Figure 1 and Figure 2). SIT and colorectal cancer expressions of miR-7, miR-21, and miR-34a were considerably different (Figure 1 and Figure 2), and these were completely different from those of SSA/P. Additionally, the expression levels of miR-31 were unchanged in all SIT cases (Figure 1 and Figure 2).

## 4. Discussion

Despite the small intestine making up 75% of the length of the digestive tract and 90% of its mucosal surface area, SIT is much rarer than colorectal and gastric adenomas and adenocarcinomas [5,6,7]. Patients with inflammatory bowel disease, mainly Crohn’s disease, and hereditary cancer syndromes such as FAP and HNPCC are at higher risk of SIT onset [5]. However, the reasons for this are not clear. The marked difference in occurrence between SIT and colorectal adenocarcinoma suggests different exposures to carcinogens [6]. This is affected by the exposure time of intestinal cells to xenobiotics or dietary carcinogens, and the time in the small intestine is shorter than in the colon due to a faster transit [6]. Furthermore, as indicated by the adenoma–carcinoma sequence, colorectal adenocarcinoma is thought to be mainly caused by colorectal adenoma [25], but the frequency of small intestinal adenoma is as low as that of small intestinal adenocarcinoma [7]. Therefore, the process of development is thought to be completely different between small intestinal and colorectal cancers. Moreover, the frequencies of APC or CDKN2A gene abnormalities in SIT are significantly lower than those in colorectal tumors [6]. In contrast, KRAS gene abnormalities are observed at the same frequency as in colorectal tumors [7].

In this study, our sample size of SIT is inadequate; it is difficult to clearly show the difference between colorectal tumors (Table 1). However, the expression profiles of miR-143, miR-145, and miR-31 are characteristic because they may clearly show the difference from colorectal tumors (Figure 1 and Figure 2). Previously, we reported that the dysregulation of miR-143 and miR-145, miR-7, miR-34a, and miR-21 might affect the endoscopic appearance and developmental pathways of colorectal tumors [18,19]. We reported that miR-143 has a significant tumor-suppressive effect on DLD-1 human colorectal cancer cells in vitro and in vivo by silencing the KRAS signaling network [26]. The expression level of miR-31 is higher in adenocarcinoma that invades the submucosa or advanced cancer and SSA/P than in carcinoma in situ or adenoma [22]. The increased expression level of miR-31 causes 5-FU resistance in colorectal adenocarcinoma by silencing FIH-1, which is associated with cancer-specific energy metabolism [22]. We reported that the downregulation of miR-143 and miR-145 makes them act as anti-oncomirs common to gastrointestinal adenocarcinomas [18,19,20,21,27].

However, even though FAP is a high-risk factor for SIT onset [5], the expression profiles of miRNAs were hardly associated with it. In addition, the expression levels of miR-31 were not upregulated in any SIT cases but were upregulated in advanced cancer and SSA/P. The serrated neoplastic pathway shows mutations in the oncogene BRAF in serrated polyps [28,29]. Moreover, high miR-31 expression is significantly associated with BRAF mutations in colorectal cancers [29]. We previously reported that BRAF mutant cells (WiDr and HT-29) show high expression of miR-31 [22]. BRAF gene abnormalities in small intestinal adenocarcinoma occur in a little less than 10% of cases, but V600E mutations account for under 1%, which is different than in colorectal cancer [6]. Most likely, the lower expression levels of miR-31 in SIT are related to the lower BRAF mutation rate.

In this study, we analyzed the relationship between SIT and miRNA profiles in comparison with colorectal tumors. Our data suggest that miR-143 and miR-145 might act as anti-oncomirs common to adenocarcinomas of the gastrointestinal tract. However, our results suggest that the miRNA expression profile of SIT is significantly different from that of colorectal tumors. In the future, we plan to further search for SIT-related miRNAs and analyze their target genes and bioinformatics analyses. We believe that these findings may contribute to useful insights for improving the diagnosis and treatment of SIT.

## Figures and Tables

**Figure 1 jcm-11-02604-f001:**
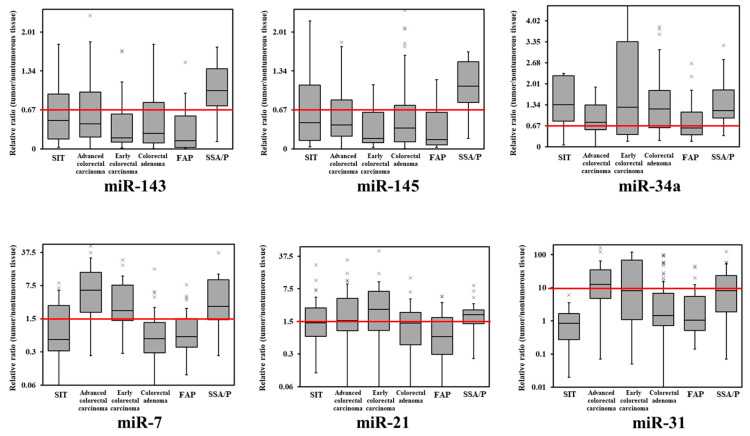
Box-and-whisker plots of miR-143, miR-145, miR-7, miR-21, miR-34a, and miR-31 expression in small intestinal tumors and colorectal tumors. The relative expression level in normal tissue was defined as 1. SIT, small intestinal tumor (adenoma and adenocarcinoma); FAP, familial adenomatous polyposis; SSA/P, sessile serrated adenoma/polyp; ×, upper outlier (over the 3rd quartile + 1.5 × interquartile range).

**Figure 2 jcm-11-02604-f002:**
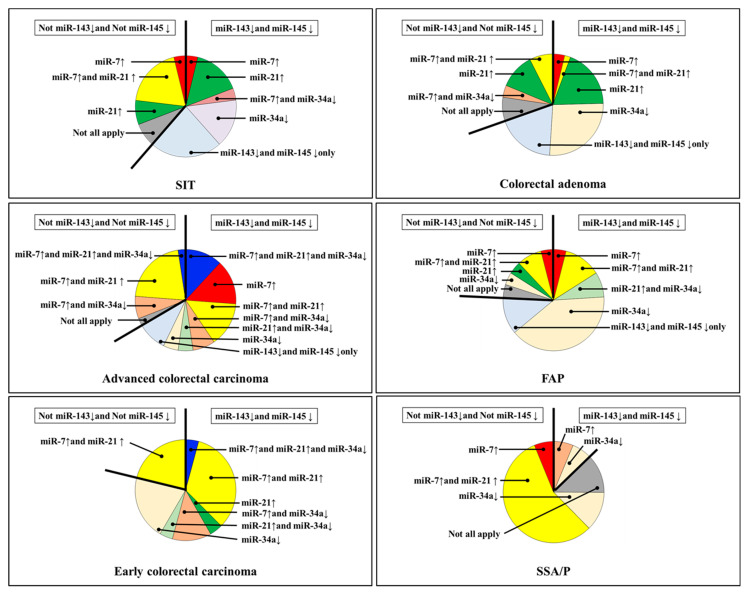
Expression profiles of miR-7, miR-34a, and miR-21 with or without downregulation of miR-143 and miR-145 in human colorectal tumors. ↓, downregulated; ↑, upregulated; SIT, small intestinal tumor (adenoma and adenocarcinoma); FAP, familial adenomatous polyposis; SSA/P, sessile serrated adenoma/polyp.

**Table 1 jcm-11-02604-t001:** Expressions of miRNA-143, miR-145, miR-7, miR-21, miR-34a, and miR-31 in small intestinal tumors and human colorectal tumor tissues were evaluated by performing TaqMan^®^ Real-time PCR assays.

tumor	n	miR-143 ↓	miR-145 ↓	miR-7 ↑	miR-21 ↑	miR-34a ↓	miR-31 ↑
SIT	27	16 (59.3%)	17 (63.0%)	8 (29.6%)	12 (44.4%)	6 (22.2%)	0 (0.0%)
advanced colorectal adenocarcinoma	46	29 (63.0%)	31 (67.4%)	37 (80.4%)	24 (52.2%)	18 (39.1%)	27 (58.7%)
*p* value (vs. SIT)		0.748	0.7	**<0.001**	0.524	0.138	**<0.001**
early colorectal adenocarcinoma	25	20 (80.0%)	19 (76.0%)	18 (72.0%)	17 (68.0%)	10 (40.0%)	10 (40.0%)
*p* value (vs. SIT)		0.093	0.309	**0.002**	0.087	0.165	**<0.001**
colorectal adenoma	59	41 (69.5%)	39 (66.1%)	12(20.3%)	25 (42.4%)	17 (28.8%)	13 (22.0%)
*p* value (vs. SIT)		0.352	0.777	0.344	0.857	0.522	**0.005**
FAP	26	20 (76.9%)	19 (73.1%)	7 (26.9%)	8 (30.8%)	14 (53.8%)	5 (19.2%)
*p* value (vs. SIT)		0.168	0.43	0.827	0.305	**0.018**	0.023
SSA/P	16	2 (12.5%)	3 (18.8%)	12 (75.0%)	11 (68.8%)	3 (18.8%)	8 (50.0%)
*p* value (vs. SIT)		**0.003**	**0.006**	**0.005**	0.109	0.554	<0.001
SIT	n	miR-143 ↓	miR-145 ↓	miR-7 ↑	miR-21 ↑	miR-34a ↓	miR-31 ↑
duodenum 1st	6	4 (66.7%)	4 (66.7%)	2 (33.3%)	2 (33.3%)	1 (16.7%)	0 (0.0%)
duodenum 2nd	12	8 (66.7%)	8 (66.7%)	2 (16.7%)	4 (33.3%)	2 (16.7%)	0 (0.0%)
duodenum 3rd	3	1 (33.3%)	2 (66.7%)	1 (33.3%)	2 (66.7%)	2 (66.7%)	0 (0.0%)
jejunum	6	3 (50.0%)	3 (50.0%)	4 (66.7%)	4 (66.7%)	1 (16.7%)	0 (0.0%)
ileum	0	-	-	-	-	-	-
*p* value		0.692	0.906	0.212	0.44	0.277	unable
SIT	n	miR-143 ↓	miR-145 ↓	miR-7 ↑	miR-21 ↑	miR-34a ↓	miR-31 ↑
adenoma	2	1 (50.0%)	1 (50.0%)	1 (50.0%)	1 (50.0%)	0 (0.0%)	0 (0.0%)
M	11	7 (63.6%)	7 (63.6%)	3 (27.3%)	4 (36.4%)	0 (0.0%)	0 (0.0%)
SM	1	1 (100.0%)	1 (100.0%)	0 (0.0%)	0 (0.0%)	1 (100.0%)	0 (0.0%)
MP	0	-	-	-	-	-	-
SS	4	1 (25.0%)	2 (50.0%)	2 (50.0%)	3 (75.0%)	2 (50.0%)	0 (0.0%)
SE	5	2 (40.0%)	2 (40.0%)	1 (20.0%)	3 (60.0%)	1 (20.0%)	0 (0.0%)
SI	4	4 (100.0%)	4 (100.0%)	1 (25.0%)	1 (25.0%)	2 (50.0%)	0 (0.0%)
*p* value		0.277	0.479	0.862	0.589	0.174	unable

A *p* value of 0.05 is considered to be significant. ↓, downregulated; ↑, upregulated; M, carcinoma in situ; SM, carcinoma invading submucosa; MP, carcinoma invading muscularis propria; SS, carcinoma invading subserosa; SE, carcinoma invading serosa; SI, carcinoma invading over serosa. SIT, small intestinal tumor (adenoma and adenocarcinoma); FAP, familial adenomatous polyposis; SSA/P, sessile serrated adenoma/polyp; bold, significant difference.

## Data Availability

The data used to support the findings of this study are available from the corresponding author on reasonable request.

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
