# Peer review of "MicroRNA Profile of Human Small Intestinal Tumors Compared to Colorectal Tumors"

_jcm, 2022, doi:10.3390/jcm11092604_

Round 1

Reviewer 1 Report

This work by Nakagawa et al aims to examine the expression profiles of a select number of miRNAs in small intestinal tumors (SIT) and other colorectal tumors to gain insight into tumor development and diagnosis. To this extent, the authors collect tumor tissues from patients and quantitate the relative expression of 6 miRNAs compared to their healthy control tissues. The authors claim that miR-143 and -145 migt act as anti-oncomirs both in adenocarcinoma of SITs and other colorectal tumors while the expression of the other 4 miRNAs was quite different.

Although the data presented could be useful to the scientists working in this field, I believe that the mere quantitation of miRNA expression level, considering the relatively small number of patients in some cancer types, is not sufficient to publish this manuscript as presented. My comments are listed below:

Major Points:

  1. Introduction, 2nd page, lines 46-47, I believe that the authors should also state the effect of miRNAs on mRNA stability, not just translation.
  2. Introduction, the authors should state how and why they selected these 6 miRNAs with some background to justify the experimental setup.
  3. Please list the primer sequences used in qPCR experiments.
  4. RNA QC or qPCR result should be included as a supplementary data to claim that FFPE qPCR results are just as reliable as those of fresh samples.
  5. Figure 1 caption, please state what each star stands for (statistically)
  6. Figure 2, I believe that the sample size is not sufficient to generate so many sub-groups for comparision. Also, the authors should make some statements about the results they present in Figure 2. Additionally, these results should be discussed in a greater detail in Discussion.
  7. It would be nice to correlate the tumor-specific expression of miRNAs to the patogenesis of each tumor type; perhaps by doing some bioinformatics analyses, e.g., potential miRNA targets and their relevance to each cancer type. Ideally, the analyses of miRNA targets would also contribute to the manuscript.

Minor Points:

  1. Abstract, line 21, “miRs-143” should be “miR-143”
  2. Introduction, 1st page, laine 44, “occur in under “ should be “occur under”
  3. The authos keep citing references 17-21 for different experiments, such as RNA isolation, qPCR, normalization..etc. Please cite the relevant reference only.

Author Response

Thank you for your review. We did additional description as your comments. Our manuscript has already been proofread by a native English speaker.

Major Points:

Comment 1

Introduction, 2nd page, lines 46-47, I believe that the authors should also state the effect of miRNAs on mRNA stability, not just translation.

Answer 1

Thank you for your comment. We did additional description on page 2, line 50- line51 and Reference 17.

Comment 2

Introduction, the authors should state how and why they selected these 6 miRNAs with some background to justify the experimental setup.

Answer 2

Thank you for your comment. The reasons for choosing the 6 miRNAs are described in Reference 18-21 (old 17-19). The reason is described in Introduction on page 2, line 51- line 54.

Comment 3

Please list the primer sequences used in qPCR experiments.

Answer 3

We performed real-time PCR using TaqMan® MicroRNA Assays (Thermo Fisher Scientific). But, Thermo Fisher Scientific does not publish the sequence of primers. We cannot describe the primer sequences.

Comment 4

RNA QC or qPCR result should be included as a supplementary data to claim that FFPE qPCR results are just as reliable as those of fresh samples.

Answer 4

Thank you for your comment. We added supplementary data as your comments on page 11.

Comment 5

Figure 1 caption, please state what each star stands for (statistically)

Answer 5

Thank you for your comment. Star is Upper outlier (Over the 3rd quartile +1.5×interquartile range). We did additional description on Legend (page 5, line 162-163).

Comment 6

Figure 2, I believe that the sample size is not sufficient to generate so many sub-groups for comparision. Also, the authors should make some statements about the results they present in Figure 2. Additionally, these results should be discussed in a greater detail in Discussion.

Answer 6

Thank you for your comment. As you pointed out, our sample size is inadequate. However, since small intestine tumors are extremely rare diseases, it was difficult to further increase the number of cases. On the other hand, the expression profiles of miR-143, miR-145, and miR-31 are characteristic, and we have determined that it is meaningful to report in J. Clin. Med. We did additional description on page 7, line 185- page 8, line 188.

Comment 7

It would be nice to correlate the tumor-specific expression of miRNAs to the patogenesis of each tumor type; perhaps by doing some bioinformatics analyses, e.g., potential miRNA targets and their relevance to each cancer type. Ideally, the analyses of miRNA targets would also contribute to the manuscript.

Answer 7

As you pointed out, these new experiments can further enhance the value of this paper. However, these studies are currently underway and no data are available. I added it to the discussion as a future task. We did additional description on page 7, line 213- line214

Minor Points:

Comment

Abstract, line 21, “miRs-143” should be “miR-143”

Introduction, 1st page, laine 44, “occur in under “ should be “occur under”

The authos keep citing references 17-21 for different experiments, such as RNA isolation, qPCR, normalization..etc. Please cite the relevant reference only.

Answer

Thank you for pointing out. We have corrected it according to your comments.

Reviewer 2 Report

  1. check the brackets in paragraph 2.1
  2. discuss Figure 2 in more detail

Author Response

Thank you for your review. We did additional description as your comments. Our manuscript has already been proofread by a native English speaker.

Comment 1

check the brackets in paragraph 2.1

Answer 1

Thank you for comment. The diagnosis of duodenal tumors was made by Esophagogastroduodenoscopy. The diagnosis of tumors in the jejunum was made with double-balloon endoscopy. Our small intestinal tumors do not include Crohn's disease and familial adenomatous polyposis.

Comment 2

discuss Figure 2 in more detail

Answer 2

Thank you for comment. Our sample size is inadequate. However, since small intestine tumors are extremely rare diseases, it was difficult to further increase the number of cases. On the other hand, the expression profiles of miR-143, miR-145, and miR-31 are characteristic, and we have determined that it is meaningful to report in J. Clin. Med. We did additional description on page 7, line 185- page 8, line 188.

Reviewer 3 Report

Interesting work but there are some points that need to be clarified

1) The authors need to better describe how were the small intestinal tumors identified. What were the symptoms that the patients had and how were the tumors diagnosed? Did they undergo capsule endoscopy, double balloon enteroscopy or both? Did the authors use push enteroscopy in some cases?

2) The authors need to better describe in the Discussion the difference between the pathways leading to the development of adenocarcinoma in the small and large bowel respectively. They mention that FAP or Crohn disease can be one of the reasons for the development of small bowel adenocarcinoma? Did their patients have FAP or Crohn's? If not were those cases sporadic adenocarcinomas of the small intestine? The authors need to further elaborate on that.

Author Response

Thank you for your review. We did additional description as your comments. Our manuscript has already been proofread by a native English speaker.

Comment 1

The authors need to better describe how were the small intestinal tumors identified. What were the symptoms that the patients had and how were the tumors diagnosed? Did they undergo capsule endoscopy, double balloon enteroscopy or both? Did the authors use push enteroscopy in some cases?

Answer 1

The first diagnosis of duodenal tumors was made by esophagogastroduodenoscopy. The first diagnosis of tumors in the jejunum was made with double-balloon endoscopy.

We did additional description on page 2, line 61-line 63.

Comment 2

The authors need to better describe in the Discussion the difference between the pathways leading to the development of adenocarcinoma in the small and large bowel respectively. They mention that FAP or Crohn disease can be one of the reasons for the development of small bowel adenocarcinoma? Did their patients have FAP or Crohn's? If not were those cases sporadic adenocarcinomas of the small intestine? The authors need to further elaborate on that.

Answer 2

Thank you for comment.

We have already described at Introduction on page 1, line 37- line 44 that the carcinogenesis of small intestine tumors and colorectal tumors is likely to be different.

Our small intestinal tumors do not include Crohn's disease and familial adenomatous polyposis. We have already described on page 2, line 61, line 69- line 71 and page 6, line 170- line 172.

Round 2

Reviewer 1 Report

The authors have nicely addressed all the concerns raised in the previous report.

Reviewer 3 Report

No more comments